
# Forecasting flood hazards in real-time: A surrogate model for hydrometeorological events in an Andean watershed

María Teresa Contreras[1,2,3], Jorge Gironás[1,2], and Cristián Escauriaza[1,2]

[1]Departamento de Ingeniería Hidráulica y Ambiental, Pontificia Universidad Católica de Chile. Av. Vicuña Mackenna 4860, 7820436, Santiago, Chile.
[2]Centro de Investigación para la Gestión Integrada de Desastres Naturales (CIGIDEN), Chile.
[3]Department of Civil and Environmental Engineering and Earth Sciences, University of Notre Dame, USA

**Correspondence:** Cristián Escauriaza (cescauri@ing.puc.cl)

**Abstract.** Growing urban development, combined with the influence of El Niño and climate change, have increased the threat of large unprecedented floods induced by extreme precipitation in populated areas near mountain regions of South America. High-fidelity numerical models with physically-based formulations can now predict inundations with a substantial level of detail for these regions, incorporating the complex morphology, and copying with insufficient data and the uncertainty posed by the variability of sediment concentrations. These simulations, however, might have large computational costs, especially if many scenarios need to be evaluated to develop early-warning systems and trigger preemptive evacuations. In this investigation we develop a surrogate model or meta-model to provide a rapid response flood prediction to extreme hydrometeorological events. We characterize the storms with a small set of parameters and use a high-fidelity model to create a database of flood propagation under different conditions. We perform an interpolation and regression procedure by using kriging on the space of parameters that characterize the events, approximating efficiently the flow depths in the urban area. This is the first application of its kind in the Andes region, which can be used to improve the prediction of flood hazards in real conditions, employing low computational resources. It also constitutes a new framework to develop early warning systems to help decision makers, managers, and city planners in mountain regions.

Keywords: Floods hazard prediction; Andean rivers; Surrogate Model; Kriging Interpolation.

## 1 Introduction

Flash floods produced by extreme precipitation events have produced devastating consequences on cities and infrastructure in mountain regions. In many cases, anthropic factors such as unplanned urban development and climate change can amplify significantly their effects, increasing the risk for the population and affecting their social and economical conditions (Wohl, 2011). The Andes mountain range in South America has been the scenario of many recent floods with catastrophic outcomes (e.g. Houston, 2005; Mena, 2015; Wilcox et al., 2016). The piedmont has also experienced a rapid urban growth, with cities





occupying regions near river channels, and increasing the exposure of the communities and their infrastructure (Barría et al., 2019). In this context, assessing flood hazards and designing strategies to reduce the potential damages caused by flooding is now critical in cities located near mountain rivers. Specifically, implementing efficient and accurate tools that facilitate real-time predictions of potential risks is a key component to provide decision-makers with enough time for action and reduce the

life-loss potential (Amadio et al., 2003; Resio et al., 2009; Toro et al., 2010; Taflanidis et al., 2013).

One of the main difficulties of predicting flood hazard in real-time in these regions comes from the very short time available to provide an accurate estimation of the risk. The complex topography in the Andean foothills is characterized by small watersheds, in the order of $50-1000$ km$^2$, and very steep slopes that can reach $40°$. As a consequence of these conditions, these types of environments often have very short concentration times (Amadio et al., 2003), which restricts the option of

real-time predictions based on instrumentation along the river. In this context, numerical models can play a significant role on understanding the propagation and consequences of floods, and on developing early warning systems for urban areas.

Implementing numerical models to simulate flash floods in these regions, however, is far from trivial, since rivers are characterized by complex bathymetries, limited field data, difficult access to obtain reliable and continuous measurements during extreme weather conditions, and large sediment concentrations (Contreras and Escauriaza, 2019). Models can provide accurate

results with high-resolution in urban areas, incorporating the coupling effect of meteorological, hydrological, hydrodynamics, and sediment transport components. Unfortunately, they might require large computational costs, with hundreds to thousands of CPU hours per simulation (Tanaka et al., 2011), which prevent the timely prediction of flood propagation that is necessary for early warning systems, especially when the uncertainty of the flood conditions requires the evaluation of multiple possible scenarios of a single event (Toro et al., 2010).

Thus, combining physical-based high-fidelity models with statistical approaches is a good alternative to provide fast response that allows the evaluation of multiple scenarios to deal with the uncertainty of future events, preserving the high-resolution and high-accuracy of physically-based hydrodynamics models. Surrogate models (or meta-models) and interpolation/regression methodologies have been already applied with the same objective for surge predictions during extratropical cyclones (Taflanidis et al., 2012, 2013). The general idea consists on developing a database of high-fidelity simulations, parameterizing each

storm/simulation through a small number of variables to represent the input for the surrogate model, and the outputs of interest. Then we provide a fast-to-compute approximation of the input/output relationship. After this surrogate model is developed, it can be used to predict scenarios for new sets of inputs or new storm characteristics that have not been simulated with the high-fidelity model.

The objective of our research is to develop a novel numerical tool for early-warning systems, based on a surrogate model

or meta-model approach, to predict efficiently the flood hazard of hydrometeorological events in an Andean watershed. We select the *Quebrada de Ramón* watershed as a study case, which is located in the Andean foothills of central Chile. We develop a set of high-fidelity simulations for a wide range of hydrometeorological scenarios by coupling two models: (1) A semi-distributed hydrological model that transforms precipitation into runoff (Ríos et al., 2019), with (2) A two-dimensional (2D) high-resolution hydrodynamic model of the non-linear shallow water equations, which incorporates the effects of high

sediment concentrations (Contreras and Escauriaza, 2019). We parameterize the set of high-fidelity simulations by selecting





parameters that describe the storm events, and one parameter that describes the flood hazard at specific points within the watershed. Then, we develop the surrogate model based on the results of the high-fidelity approach. This surrogate approach can rapidly calculate new scenarios defined by specific values of the 4 parameters that describe an event, using a statistical interpolation on the parameter space. The high-fidelity model can therefore capture the dynamic interplay between the high

sediment concentrations and geomorphic drivers on the flood propagation in mountain streams, and reveal the most important physical aspects of the flow (Contreras and Escauriaza, 2019). The surrogate model, on the other hand, provides high-resolution results with low computational costs, considerably more inexpensive compared to traditional computational fluid dynamics (CFD) simulations (Couckuyt et al., 2014; Jia et al., 2015), by using a database of pre-computed cases for the fast evaluation of multiple scenarios, which can be employed to develop early warning systems. In this work we implement Kriging or Gaussian

Process interpolation, a popular surrogate modeling technique to approximate deterministic data (Couckuyt et al., 2014).

The paper is organized as follows: In section 2 we provide a detailed description of the generation of the database that we use to build the new surrogate approach, using high-fidelity simulations that combine a hydrological model and a 2D hydrodynamic model for the *Quebrada de Ramón* stream near Santiago, Chile. In Section 3 we describe the statistical interpolation of the surrogate model using kriging. Results for water depth predictions in critical points and flooded areas are described in section

4. In section 5 we provide a discussion of the results, and in the conclusions of Section 6 we summarize the findings of this investigation and discuss future research directions.

## 2   Database Generation though High-fidelity Simulations

The first requirement for the implementation of the surrogate model is the development of a database of high-fidelity simulations that can describe the complex relationships between meteorological, hydrological, and hydrodynamic processes during

the storms. For this purpose, we select the *Quebrada de Ramón* watershed as our case of study (Contreras and Escauriaza, 2019). This basin is located in central Chile, to the east of the city of Santiago. The record of flash floods shows that extreme precipitation is generally accompanied by an increment of temperature and a higher elevation of the 0°C isotherm during events affected by El Niño phenomenon.

The main channel experiences an elevation change in the range from 3400 to 800 m asl, and the total area of the watershed

is 38.5 km$^2$, which generates steep gradients that produce high flow velocities and transport large sediment concentrations to the urbanized areas. The flood risk has increased over the last years, since in the lower section of the watershed, the natural channel has been modified and diverted into a concrete channel with a design discharge of 20 m$^3$/s, which does not exceed the 10-year return period (Catalán, 2013).

To create the database we couple hydrological and hydrodynamic models to run 200 years of continuous synthetic precipita-

tion and temperature series, simulating the rainfall-runoff process in the watershed. We use a hydrological model described in section 2.1 to compute the time series of river discharge at the outlet of the natural domain, which corresponds to the entrance to the urbanized area. Then, we select events such that the maximum instantaneous flow rate is equal or larger than 20 m$^3$/s, obtaining a set of 49 different hydrographs. We use the hydrodynamic model of Contreras and Escauriaza (2019), which we



describe briefly in section 2.2, to propagate these hydrographs through the urban area for different sediment concentrations, which are quasi-randomly selected within a realistic range for the region. Thus, each simulation in the database describes the flow depths and velocities during the flood in the study area, produced by specific hydrometeorological conditions and the high-resolution topography of the terrain.

In the remainder of this section, we provide a detailed description of both, the physics-based hydrological and hydrodynamic models, and their implementation in the Andean watershed.

## 2.1 Hydrological Model

We develop a continuous rainfall-runoff model using Hec-HMS, which transforms meteorological data into discharge hydrographs, based on the work of Ríos et al. (2019). We divide the watershed in subcatchments based on a digital elevation model
(DEM) and we calculate the abstractions for each of the subcatchments (i.e. interception, surface storage and infiltration), the evapotranspiration, and the snow melting and accumulation. We then compute the base flow and runoff, and propagate the hydrographs to the outlet of the watershed.

To determine the effective precipitation we estimate the abstractions, we use a simple vegetation interception model that considers a maximum storage capacity for each subcatchment. This function depends on the type of vegetation, the season of the year, and the characteristics of the storm (Ponce, 1989). We use a similar method to define the surface storage, which
depends on the soil properties and the slope (Bennett, 1998). Both of these variables must be filled before the initiation of infiltration and/or surface runoff. We use the Soil Moisture Accounting method (SMA) (Scharffenberg and Fleming, 2010) to simulate infiltration, after the processes of interception and surface storage. The SMA method represents the catchment as a group of different subsurface storage strata, in which we simulate infiltration, percolation, the soil humidity dynamics, and the evapotranspiration continuously during wet and dry periods. To obtain the continuous modeling of potential evapotranspiration,
we use the Priesley-Taylor equation, which integrates the effects of the net solar radiation, the regional advection, and the air temperature (Chow et al., 1988). The snow accumulation and melting is simulated as a function of the atmospheric conditions through the temperature index (Bras, 1990). In this hydrological model, the melted snow becomes available on the soil surface and then it is added to the hyetograph of the catchment (see Scharffenberg and Fleming, 2010, for details).

To estimate the base flow in the subcatchments we use of a linear reservoir model with an exponential recession curve.
We compute the runoff with the Clark Unit-Hydrograph, which propagates the time-land curves through the linear reservoir to obtain the response of the catchment (see Viessman and Lewis, 2003, for details). Finally, due to the low storage and attenuation of floods in these environments, we use a simple delay method without attenuation to simulate the transit of the discharges from every subcatchment to the outlet.

For the Quebrada de Ramón, we divide the watershed in 12 subcatchments, and obtain their morphological metrics from a 2
m resolution digital elevation model (DEM), as shown in Figure 1. We use daily precipitation and temperature data recorded at Quinta Normal station, located at 527 m asl, at around 10 km West of the outlet of the watershed. The data comprises the period from the 1st of April of 1971 to the 31st of March of 2010, which represents 40 hydrological years. To perform the simulation, we statistically disaggregated the daily record of precipitation in time to generate an hourly synthetic record for a total of 200





years. We use the method of Ríos et al. (2019), which is a modification of the method developed by Socolofsky et al. (2001). The hourly disaggregated rainfall records to the different subcatchments is extrapolated considering the elevations of their centroids. We compute the annual precipitation between 1995 and 2014 for 23 stations located in a surrounded area of 3000 km$^2$, whose elevations vary between 176 and 2475 m asl, and calculate the mean spatial gradient of the annual precipitation

for the set of stations, 126 mm/km, to extrapolate the hourly record of precipitation from the elevation of the Quinta Normal station to the elevation of the centroid of each subcatchment.

Likewise, we extrapolate the records of maximum and minimum daily temperature of Quinta Normal to each subcatchment following the $-6.5$ °C/km gradient, typically used in the zone (Torrealba and Nazarala, 1983; Lundquist and Cayan, 2007) The daily record is disaggregated in an hourly time series, using a procedure based on a Fourier decomposition (Campbell

and Norman, 2012). Since there are no series of hourly solar radiation data in the study area, we use the monthly average of solar radiation in the period $2003 - 2012$ for the entire watershed, which are very similar for the region (Ministerio de Energía, 2015).

Due to the lack of a long and reliable record of runoff data at Quebrada de Ramón, we calibrate the model by replicating the annual flood frequency curve. We calibrated the model by following the methodology presented in Ríos et al. (2019). We run

the model during the 200 years of precipitation and temperature record, which are generated after the temporal desegregation of the daily series, with a time step of 30 minutes.

In Figure 2, we depict the frequency curves of annual maximum discharges and then compare the results with the most recent curve reported in the study area (ARRAU Ingeniería, 2015), which is also consistent with previous studies. For additional details about the implementation of the hydrological model, the reader is referred to Ríos et al. (2019).

## 2.2  Hydrodynamic Model

We propagate the floods by solving the non-linear shallow-water equations (NSWE), which correspond to the conservation of mass and momentum, assuming hydrostatic pressure distribution, negligible vertical velocities, and vertically-averaged horizontal velocities. Since the high sediment concentrations can change the rheology of the flow, we modify the equations to account for the heterogeneous density distribution (Contreras and Escauriaza, 2019). We couple the sediment mass conservation

equation, and include an additional source term in the momentum equations to represent the rheology of the mixture of water and sediments. In the numerical solution, we compute the evolution in time and space of the flow depth $h$, the volumetric sediment concentration $C$, and the cartesian components of the 2D flow $u$ and $v$ in the $X$ and $Y$ directions, respectively.

The equations are solved in non-dimensional form, using a characteristic velocity scale $\mathcal{U}$, a scale for the water depth $\mathcal{H}$, and a horizontal length scale of the flow $\mathcal{L}$. Two non-dimensional parameters appear in the equations: (1) The relative density

between the sediment and water $s = \rho_s/\rho_w$; and (2) The Froude number of the flow $Fr = \mathcal{U}/\sqrt{g\mathcal{H}}$.

To adapt the computational domain to the complex arbitrary topography in mountainous watersheds, we use a boundary-fitted curvilinear coordinate system, denoted by the coordinates $(\xi, \eta)$. Through this transformation, we can have a better resolution in zones of interest and an accurate representation of the boundaries. We perform a partial transformation of the

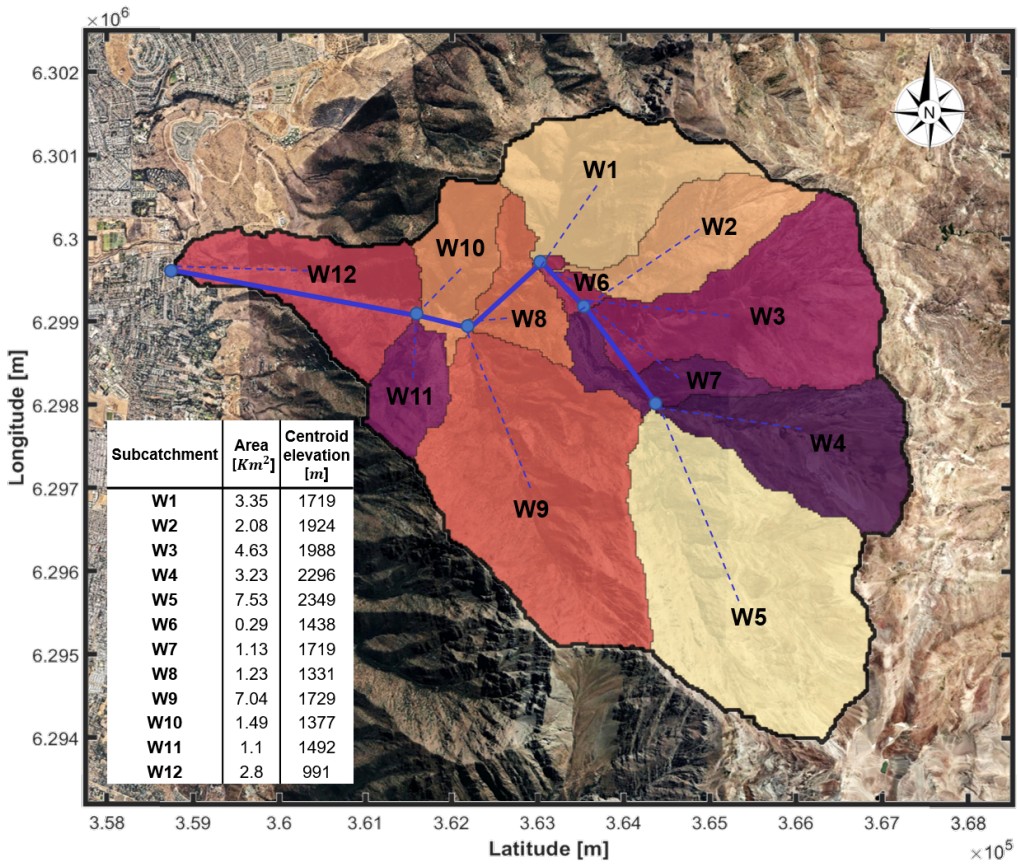

**Figure 1.** Configuration of the Quebrada de Ramón used in the hydrological model. The colored areas represent each of the subcatchments, enumerated from 1 to 12 in Hec-HMS. The blue lines represent each of the reaches, and the blue dots the junctions between them (Background image from ⓒ Google Earth).

equations, and write the set of dimensionless equations in vector form as follows,

$$\frac{\partial Q}{\partial t} + J\frac{\partial F}{\partial \xi} + J\frac{\partial G}{\partial \eta} = S_b(Q) + S_S(Q) + S_C(Q) \tag{1}$$

where $Q$ is the vector that contains the non-dimensional cartesian components of the conservative variables $h$, $hu$, $hv$ and $hC$. The Jacobian of the coordinate transformation $J$ is expressed in terms of the metrics $\xi_x$, $\xi_y$, $\eta_x$ and $\eta_y$, such that $J = \xi_x\eta_y - \xi_y\eta_x$.

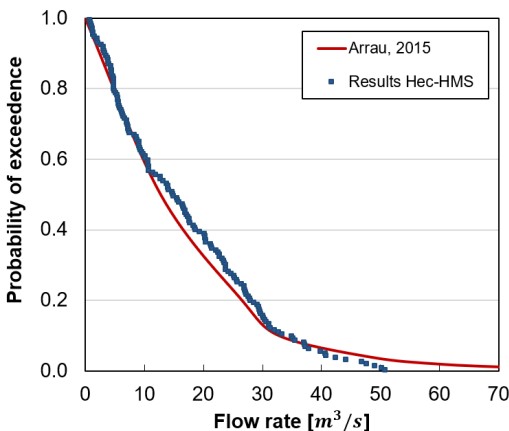

**Figure 2.** Calibration of HEC-HMS: Comparison between measured flow rate and results from the hydrological model.

The fluxes $F$ and $G$ in each coordinate direction expressed as follows,

$$F = \frac{1}{J} \begin{pmatrix} hU^1 \\ uhU^1 + \frac{1}{2Fr^2}h^2\xi_x \\ vhU^1 + \frac{1}{2Fr^2}h^2\xi_y \\ ChU^1 \end{pmatrix}, G = \frac{1}{J} \begin{pmatrix} hU^2 \\ uhU^2 + \frac{1}{2Fr^2}h^2\eta_x \\ vhU^2 + \frac{1}{2Fr^2}h^2\eta_y \\ ChU^2 \end{pmatrix} \tag{2}$$

where $U^1$ y $U^2$ represent the contravariant velocity components defined as $U^1 = u\xi_x + v\xi_y$ and $U^2 = u\eta_x + v\eta_y$, respectively.

The model considers three source terms: $S_B$ contains the bed slope terms, $S_S$ corresponds to the bed, and internal stresses of the flow and $S_C$ incorporates the effects of gradients of sediment concentration.

To account the rheological effects, in the $S_S$ term, we modify the quadratic model of O'Brien and Julien (1985) to represent the stresses for a wide range of sediment concentrations, expressing clearly the contribution of each physical mechanism. Thus, the source terms for the bed and internal stresses for the $i$ coordinate direction can be written as:

$$S_{Si} = S_{yield} + S_{v_i} + S_{td_i} \tag{3}$$

where $S_{yield}$ represents the sum of the yield and Mohr-Coulomb stress, $S_{v_i}$ the viscous stress and $S_{td_i}$ the sum of the dispersive and turbulent stresses. We modify the model changing the equations to represent each of the terms from the originally used, which have been obtained from experiments or physically-based formulas, to the traditional flow resistance formulas that converge to zero when the concentration is zero.

The system of equations is solved in a finite volume scheme based on Guerra et al. (2014), which has shown high efficiency and precision to simulate extreme flows and rapid flooding over natural terrains and complex geometries. The method is implemented in two steps: First, in the so-called hyperbolic step, the Riemann problem is solved at each element of the discretization without considering momentum sinks. The flow is reconstructed hydrostatically from the bed slope source-term,





adding the effects of the spatial concentration gradients. In the second step, we incorporate the shear stress source terms utilizing a semi-implicit scheme, correcting the predicted values of the hydrodynamic variables. To compute the numerical fluxes, we implement the VFRoe-NCV method, linearizing the Riemann problem (Guerra et al., 2014). The MUSCL scheme is used to perform the extrapolation with second-order accuracy in space. The model has been validated for many cases including

supercritical flows and wave propagation on dry surfaces, also comparing the results with analytical solutions and experiments that include sharp density gradients (for additional details of the model, the reader is referred to Contreras and Escauriaza, 2019).

For the Quebrada de Ramón watershed, we simulate the propagation of the 43 hydrographs obtained from the hydrological model, whose maximum flow rate exceeds 20 m$^3$/s, which is the maximum hydraulic capacity at the critical point in the

urbanized area.

We define the computational domain shown in Figure 3 that comprises the channelized portion of the main channel (6.6 km approximately), which starts at an elevation of 878.8 m asl. A curvilinear boundary-fitted grid is used to perform the simulations, consisting of a total of $3,914 \times 697$ grid nodes. The grid resolution varies progressively in the flow direction from 0.5 m upstream, to 2 m of resolution within the flooding zone. On the cross-stream direction, the resolution of the grid varies

from approximately 0.5 m near the main channel to more than 30 m in areas that are never flooded. To construct the grid, we couple a 1 m resolution LIDAR survey of the area around the channel, with 0.5 m resolution topographic field measurements to define the main channel and a 0.5 m digital surface model (DSM) from satellite images for the rest of the watershed.

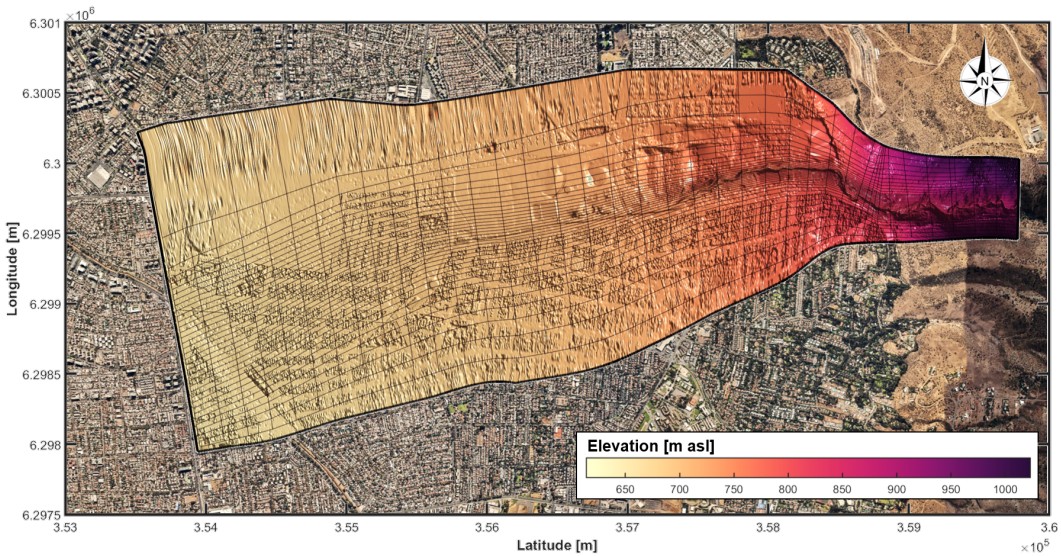

**Figure 3.** Aerial view of the urbanized area of the Quebrada de Ramón watershed. The area enclosed by the black line defines the computational domain used in the hydrodynamics model. The grid is presented every 100 nodes in the direction of the flow, and every 20 nodes across the river channel. Background image from © Google Earth.





The bed roughness is represented by a mean sediment grain diameter $d_s$, which is shown in Figure 4. We use field measurements at the locations shown in the orange symbols to determine $d_s$ along the channel. We derived values of $d_s$ from Manning coefficients in the floodplain, by using the Stricker relation (Julien, 2010). Based on satellite images and field observations, we define two types of land cover: Floodplain with short grass ($n = 0.025$) in parks, and rough asphalt ($n = 0.016$) in the rest of the urbanized area.

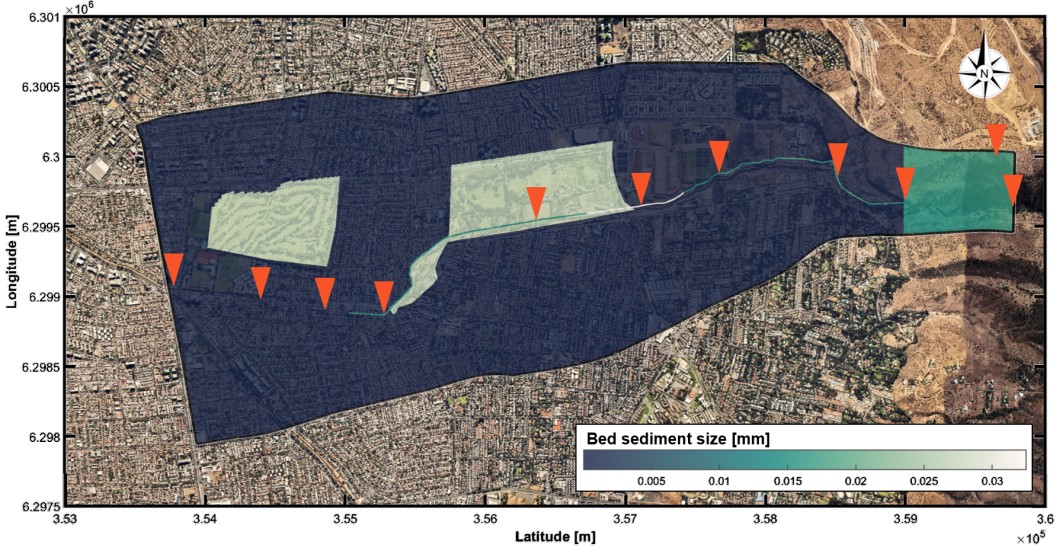

**Figure 4.** Distribution of the mean sediment size in millimeters. Orange triangles denote the sites where we perform direct measurements of sediment size distributions. Background image from © Google Earth.

The hydrographs resulting from the hydrological model correspond to the inflow boundary condition in the eastward boundary, and open boundary conditions are defined at all the other boundaries of the computational domain. The initial condition for all the cases is dry-bed since we are interested in simulating the propagation of single events, and the channel is typically dry before these events. We run the simulations for a total physical time equivalent to the length of the hydrograph plus a concentration time of the watershed ($\sim 3$ hours). We use a simulation time step defined by the CFL stability criterion, with values within $0.4 - 0.6$.

Due to the lack of measurements of sediment concentrations during floods, the values of concentration for each simulation are defined by using a Latin hypercube sampling in the range of concentrations $0$ to $40\%$, which have shown to be the most significant changes on the flood propagation (Contreras and Escauriaza, 2019).

Figure 5 shows the instantaneous results of the simulations for a peak flow of 97 m³/s with $38\%$ of sediment concentration. The resolution of the simulations in time and space shows the evolution of the flow in the urban area and the zones where the sediment concentration produces significant changes on the velocity and flow depth of the flood.


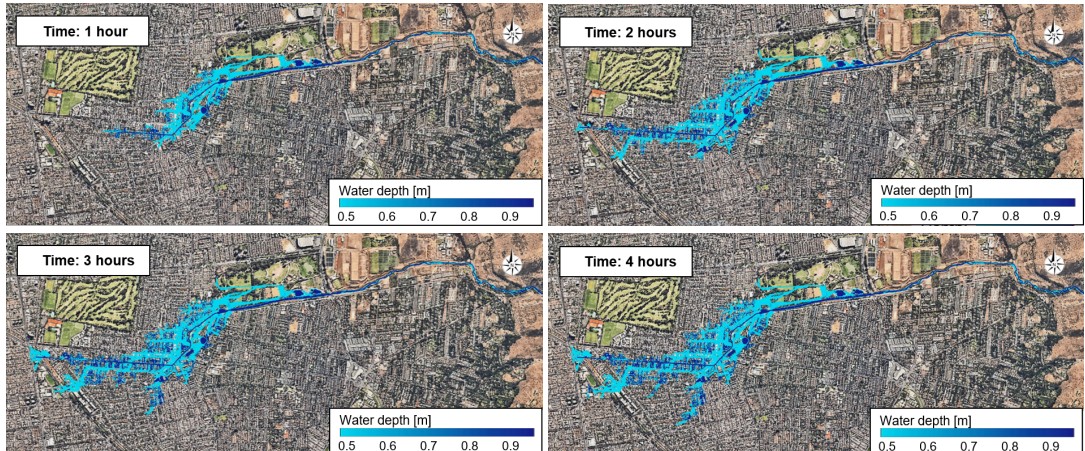

**Figure 5.** Instantaneous flow depth in the *Quebrada de Ramón* and the flooded area in Santiago, Chile, for a peak flow of 97 m$^3$/s and 38% of sediment concentration. Background image from © Google Earth.

## 2.3 Definition of Inputs and Outputs of the Surrogate Model

Parameterizing the database and defining a small number of input and output variables that can capture the characteristics of the storms and the flood propagation requires an understanding of the physics of the flood generation. In this section we provide a brief overview of the main physical components that control flooding in Andean environments, and define the parameters used
to implement the surrogate model.

From the hydrometeorological perspective, the volume of precipitation and its spatial and temporal distribution are the critical factors that define the magnitude of the flood. Most of the precipitation in the region of study occur between May to September, which constitutes the rainy season. Storms are typically frontal rainfall events that last from 1 to 2 days (Garreaud, 1993). They correspond to 80% of the total events in the area, accumulating 50% of the annual precipitation (Montecinos,
1998). Cyclonic events, however, also take place in the mountains, producing storms with intense precipitation. While these events occur occasionally in the area, they can produce extreme flash flooding. El Niño Southern-Oscillation (ENSO) is one of the factors that also generates floods in the Andes (García, 2000; Sepúlveda et al., 2006). This phenomenon cyclically produces years of severe drought or frequent floods (Rutllant et al., 2004). During rainy years, the intensity and duration of the storms is larger, often producing flash floods and debris flows in the watersheds along the Andean range.
Warm temperatures during storms, especially when they are influenced by the ENSO phenomenon, are a critical factor that might increase the flooding effects (Vicuña et al., 2013). Due to the complex topography in these environments, watersheds are very susceptible to the variation of the elevation of the 0°C isotherm. The warm conditions expand the total contributing area of liquid precipitation, increasing the runoff volume, the peak discharges, and producing high flow velocities and sediment transport in the flow.





Steep topographic gradients of the river channels promote the mobilization of large amounts of sediments, mostly from landslides. In many cases, the volumetric sediment concentrations in the flow can exceed 20%, which requires to consider the additional stresses produced by the particle-flow and particle-particle interactions (Julien, 2010). In those cases, the rheology of the flow changes from the traditional resistance term that only considers the bed stresses.

The changes on the flow hydrodynamics due to high sediment concentrations are also evident on the flood propagation. In the study region, Contreras and Escauriaza (2019) showed that the flooded area in the urban zone increases 36%, if we consider the same flood with clear water or with a concentration of 20%. Likewise, the water depth in the urban area can increase 25% when 60% of sediment concentration is considered instead of clear water.

In this framework, we characterize the extreme events in the watershed by only four parameters to define the intensity storms, and implement the surrogate model. This simplified description of floods scenarios considers a small number of model parameters based on the storm characteristics, incorporating also the effects of the sediment concentration. These parameters are the following:

1. Mean value of the precipitation event $\overline{P}$, which contains information of the intensity of rainfall;

2. The second statistical moment of the precipitation $M_{2_P}$, to incorporate the temporal distribution and duration of the event;

3. The minimum temperature of the event $T_{Min}$, which yields information on the elevation of the 0°C isotherm; and

4. Sediment concentration (C), to consider the volumetric and rheological effects of the flood, i.e., the changes of velocity and depth, especially when we have a hyperconcentrated flow.

The output of the surrogate model can be defined as almost any variable derived from the time series of water depth or flow velocity distributed in space. Here we select the maximum water depth at the location where the channel reduces its hydraulic capacity to 20 m$^3$/s. Thus, the vector of inputs $X$ and $Y$ that we use to implement the surrogate model are encoded as follows,

$$X = \begin{bmatrix} \overline{P} & M_{2_P} & T_{Min} & C \end{bmatrix}; \qquad Y = [h_{Max}] \tag{4}$$

In the next section we explain the statistical procedure to create new scenarios from the interpolation of pre-computed cases of the database.

## 3  Surrogate Model: Kriging on the Parameter Space

After constructing the database with 49 simulations of different storms, and parameterizing them by defining the inputs and the output of the simulations, we can implement the surrogate model and define the simplified relationship among the variables. In this case we use the DACE procedure, which has been designed to implement a variety of Kriging formulations and extensions for surrogate formulations (Couckuyt et al., 2014).





In general terms, Kriging considers the response of the model $\hat{Y}(X)$ to a vector of inputs $X = \left[\mathbf{x}^1,\ \mathbf{x}^2,\ \mathbf{x}^3,\ ...,\ \mathbf{x}^d\right]$ as a linear combination or regression function $f(X)$ that captures the general trend of the data, and a random function $R(X)$ that describes the residuals of the stochastic process (Lophaven et al., 2002):

$$\hat{Y}(X) = f(X) + R(X) \tag{5}$$

Defining the correct regression function is not a simple task, since this function should capture the complexity in the input/output relationship. The DACE approach provides several options, and for simplicity, the most used ones are the *simple* and the *ordinary* Kriging, in which the regression function is defined as a zero-order or constant function, respectively. In this investigation we use *universal* Kriging, which provides more flexibility to the regression function and defines $f(X)$ as a linear combination of $p$ chosen functions, as follows:

$$f(X) = \sum_{i=1}^{p} \alpha_i b_i(X) \tag{6}$$

where $b_i(X)$ are the basis functions, and $\alpha_i$ are arbitrary coefficients. Assuming that the values of $\alpha_i$ are correctly determined by Generalized Least Squares (GLS), the random function $R(X)$ becomes white noise, and it can be represented by a Gaussian Process $Z(X)$ with mean 0, variance $\sigma^2$, and a correlation matrix $\Psi$ (Lophaven et al., 2002). Thus, the formulation of the surrogate model can be written as,

$$\hat{Y}(X) = f(X) + Z(X) \tag{7}$$

The implementation of the surrogate model requires a matrix with $n$ samples $S = [\mathbf{s}_1,\ \mathbf{s}_2,\ \mathbf{s}_3,\ ...,\ \mathbf{s}_n]^T$, with $\mathbf{s}_i \in \mathrm{IR}^d$, and its respective deterministic or high-fidelity responses $Y = [\mathbf{y}_1,\ \mathbf{y}_2,\ \mathbf{y}_3,\ ...,\ \mathbf{y}_n]^T$ with $\mathbf{y}_i \in \mathrm{IR}^q$. To determine the functions $b_i(X)$ in the regression function, we assume that they are power-based polynomials of degree 0, 1, or 2, and that they are encoded in the matrix $F(i,j) = b_j(\mathbf{s}_i)$ with $i \in \{1,n\}$, $j \in \{1,p\}$.

On the other hand, the Gaussian process $Z(X)$ is defined by the correlation matrix $\Psi(i,j) = \psi(\mathbf{s}_i,\mathbf{s}_j)$ with $(i,j) \in \{1,n\}$. Each term of the matrix is computed from a stationary correlation function (Couckuyt et al., 2013):

$$\psi(\mathbf{s}_i,\mathbf{s}_j) = \exp\left(-\sum_{a=1}^{d} \theta_a |\mathbf{s}_i(a) - \mathbf{s}_j(a)|^k\right) \tag{8}$$

where $\theta_a$ and $k$ are parameters that can be fixed or determined by using a Maximum Likelihood Estimation (MLE) depending on the correlation function chosen. The DACE approach can consider multiple options for the correlation functions, for instance

Gaussian, Exponential, Cubic, Linear, Spherical, Spline, Matérn-3/2, or Matérn-5/2.

The mean and variance of the prediction are estimated as:

$$\hat{Y}(X) = M\alpha + r(X)\Psi^{-1}(Y - F\alpha), \qquad s^2 = \sigma^2\left(1 - r(X)\Psi^{-1}r(X)^T + \frac{1 - F^T\Psi^{-1}r(X)^T}{F^T\Psi^{-1}F}\right) \tag{9}$$

where $M$ is the matrix $F$ of basis functions evaluated in the input vector $X$, and $r(X)$ is a vector of correlations $r(X) = [\psi(X,\mathbf{s}_1),\ \psi(X,\mathbf{s}_2),\ ...,\ \psi(X,\mathbf{s}_n)]$. Lophaven et al. (2002) and Couckuyt et al. (2013) provide a detailed description of the

*universal* Kriging in DACE, which is implemented here.





# 4   Results

## 4.1   Cross-validation process: Implementing the Surrogate Model

Determining the correct order of the regression function and the correlation function is critical to optimize the accuracy of the
surrogate model. Since there is no clear methodology to define them, we implement a cross-validation process to assess the
accuracy of a surrogate model built with multiple combinations of regression and correlation functions.

Traditionally, the cross-validation process consists of dividing the database into two groups. The first set is used to implement
the surrogate model, and the second one to validate the results and evaluate the accuracy of the predictions. In this research,
reducing the number of storms in the database might significantly affect the accuracy of the model, since the total number of
simulations is not large enough. To deal with this setback, we run the cross-validation process by selecting one testing storm at
a time, and building the surrogate model with the remaining 48 storms. We carry out this process by taking out of the database
the 49 storms, one at a time.

We evaluate the performance of the model based on the mean square error (MSE), and the percent of cases in which the
high-fidelity value is within one standard deviation from the mean of the prediction (PWSD), for values of water depth at the
point in the urban area where the hydraulic capacity of the channel reduces to 20 m$^3$/s. We tested three orders of the regression
function (Order 0, 1, and 2), and the nine correlation functions available in DACE (Cubic, Exponential, Gauss, Gauss-p, Linear,
Matérn-3/2, Matérn-5/2, Spherical, and Spline).

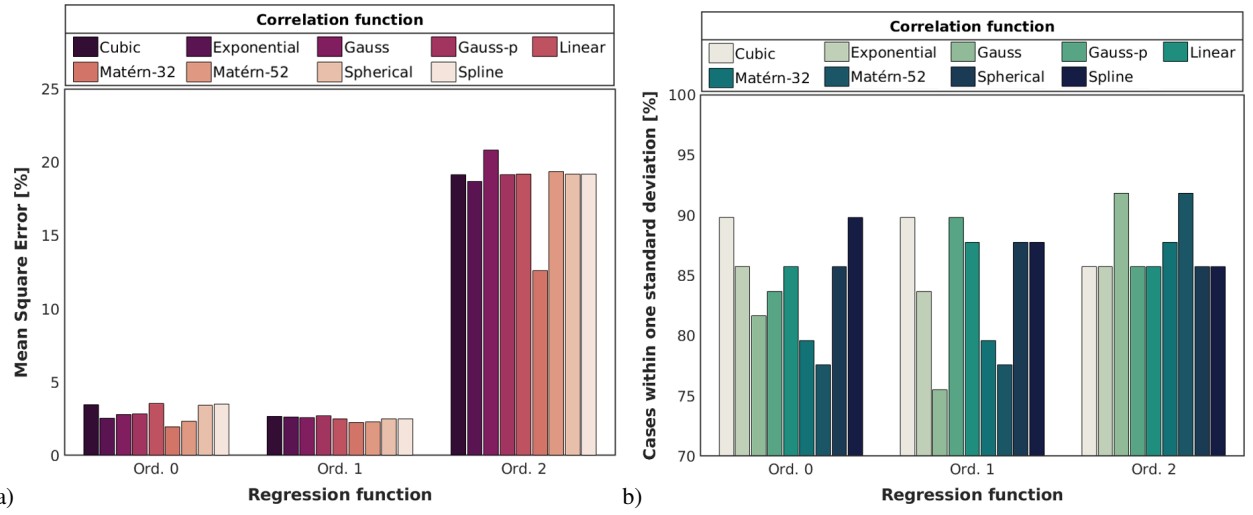

**Figure 6.** Assessment of the surrogate model built with all the available combinations of regression and correlation functions. Figure a)
shows the performance in terms of the Mean Square Error, and b) the percentage of cases predicted within one standard deviation.

Figure 6 shows the performance for the 27 different surrogate models. The MSE, on the left, is significantly reduced when
we use a zero or first order regression function, in comparison with the second order one, but it does not strongly depend on the
correlation function. On average, the MSE for the second order regression function is 18.59%, and it decays to 2.92% and 2.51%




for a zero and first order function, respectively. In terms of the percentage of cases predicted within one standard deviation (PWSD), the performance of the different regression functions is associated with the correlation function used compared to the order selected for the regression function. For the case of a zero order function, we observe the worst performance of the model for a Matérn-5/2 (PWSD=77%), and the best one for a Cubic or Spline function (PWSD=90%). For a first-order function,

the Gauss and the Cubic or Gauss-p yield the best and the worst results, respectively, with PWSD=75% and 90%. For the second-order function, in general, the value of PWSD is larger for all the correlation functions. The worst cases get over 85%, and the best ones, the Gauss and Matérn-5/2 up to 90%.

Based on the significant differences of MSE with the regression functions, we decide to implement the surrogate model with the best combination of MSE and PWSD, but only considering the zero and first order regression functions. We notice that the

linear correlation function reaches the minimum MSE for both orders of regression functions. Besides, this gets up to one of the highest PWSD, only surpassed by the Cubic and the Gauss-p functions for 2.5%.

Thus, we use a surrogate model built with a first order regression function and a linear correlation function from this point forward. The next subsection shows how this surrogate model performs in terms of the relative and absolute error, and how the different input parameters affect the accuracy of the predictions of water depth at the specific critical point in the urban area.

Then, we explore the idea of a surrogate model distributed in space to predict flooded areas.

## 4.2    Prediction of Maximum Water Depth at the Critical Point

We study the performance of the surrogate model built with a first order regression function and a linear correlation function for specific events. We use the same methodology to run the cross-validation process in the previous subsection and examine the absolute and relative error of the prediction. We aim at understanding when the surrogate model performs the best and the

poorest depending on the characteristics of the storms.

Figure 7a) shows in red the absolute error between the high-fidelity value and the mean of the prediction for each of the 49 storms. The mean of the error is 0.13 m, with a minimum of only 0.0064 m (Ev. 27) and a maximum that raises to 0.73 m (Ev. 1 and 48). The majority of the events have an error equal or smaller than 0.25 m (44 storms or 89.8%), 6.12% is in the range 0.25-0.5 m, and only two of the events, a 4%, have an error that exceeds 0.5 m. The blue bars represent one standard deviation

of the prediction plotted in the positive and negative directions around zero. Their values vary from a minimum of 18.24 cm to a maximum of 82.78 cm, with a mean of 29.21 cm. Even though 87.76% of the predictions are within one standard deviation, we observe that there is an association between the high standard deviations and larger errors.

Based on these results, we plot in Figure 7b) the relative error for each of the events, and highlight with red those with a standard deviation higher than 10% of the mean of the maximum water depth at the point of prediction, for the total 49 storms.

The mean of the relative error for all the events is 3.6%, with a minimum and maximum equal to 0.18% and 19.92% respectively. However, if we consider a valid prediction as those with $s \leq 10\% \overline{h_{HF}}$, we can establish that the surrogate model can predict 87.76% of the storms, with a mean, minimum and maximum relative error equal to 2.22, 0.18, and 6.24% respectively.

We also focus on understanding how the surrogate performs depending on the characteristics of the storms. In Figure 8, we plot the relative error as a function of the four inputs parameters $X = \begin{bmatrix} \overline{P} \ M_{2_P} \ T_{Min} \ C \end{bmatrix}$ for each of the 49 storms.



Natural Hazards
and Earth System
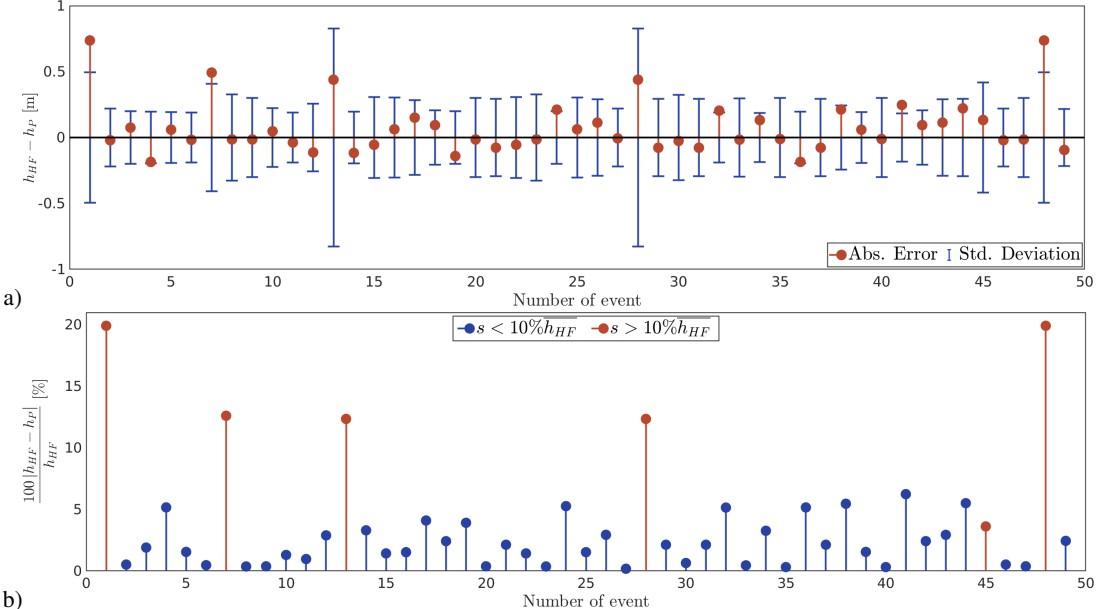

**Figure 7.** Results of the surrogate model for the water depth, compared to the data obtained from the high-fidelity approach. Figure a) shows in red the absolute error between the high-fidelity value and the mean of the prediction, and in blue one standard deviation of the prediction around zero. Figure b) shows the relative error of the model. The events in red represent those with a predicted standard deviation larger than 10% of the mean maximum water depth at the point of the predictions for the entire set of high-fidelity values $\overline{h_{HF}}$.

Figure 8a) shows that the two predictions with more than 10% of relative error are events with $\overline{P} > 2$ mm. This might show that the surrogate model losses accuracy for intense storms; however, the model is also able to predict storms with $\overline{P} > 3$ mm and errors that do not exceed 5%.

We show the magnitude of $M_{2_P}$ in Figure 8b), as a representation of the distribution of precipitation for the total duration

of the storms. The performance of the model does not depend strongly on the values of the second moment of precipitation. The two predictions with relative errors that become larger than 10% are storms with $M_{2_P}$ in the order of 6 mm$^2$, over the average of the parameter in the database. Nevertheless, four other predictions with a similar value of $M_{2_P}$, and three others with $M_{2_P} > 8$ have relative errors that do not exceed 5%.

Another important factor that influences the magnitude of the flood is the minimum temperature during the storms. In Figure

8c), we show that the surrogate model exhibits good performance at predicting storms with minimum temperatures that exceed 10° C, with relatives errors lower than 5%. Indeed, the two storms with the largest relative error ($\sim 20\%$) occur under low temperatures ($<5°$ C).

The sediment concentration, shown in Figure 8c), does not have a specific relationship with the magnitude of the relative error either, since the predictions with errors that are larger 10% take place in events with sediment concentrations in almost

the entire range of possible values ($0 - 40\%$). Therefore, we do not observe that the surrogate model does especially good or bad for specific storms with specific characteristics. We consider this aspect as a useful property of the model. Having a lightly


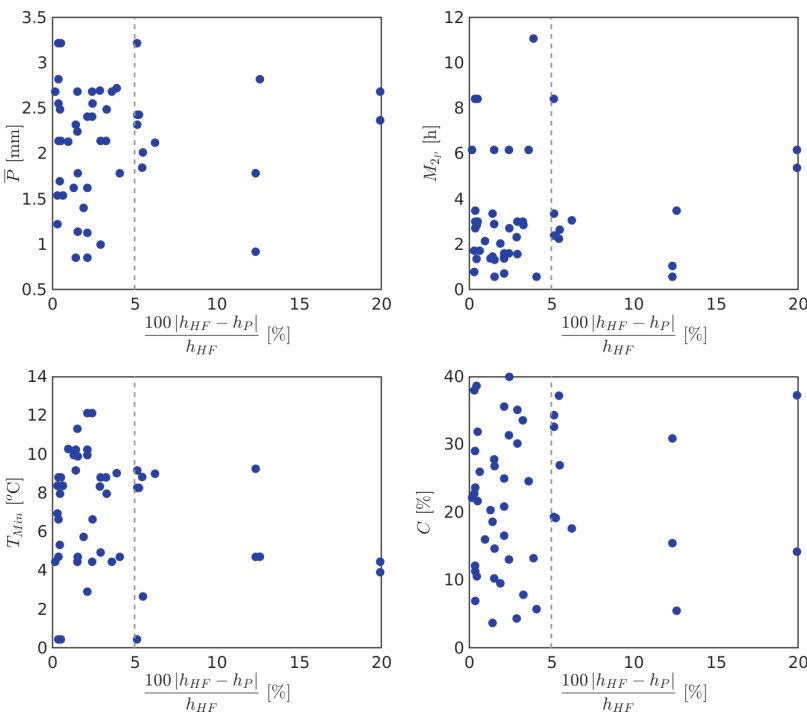

**Figure 8.** Results of the surrogate model for the four input parameters ($P$, $M_{2_P}$, $T_{Min}$, and $C$), compared to the data obtained from the high-fidelity approach.

uniform distribution of cases in the database is highly-recommended, this property of the model allow us to predict events that do not occur very often without increasing significantly the errors of the estimation.

We also study the relationship between the accuracy of the model, and the magnitude of the flood, through the maximum discharge at the prediction point, $Q_{Max}$, as shown in Figure 9. While the range of $Q_{Max}$ in the database varies between 20

5    to 46 m³/s, the predictions with relative errors greater than 10% are floods with $Q_{Max} <= 40$ m³/s. More specifically, the two cases with error $\sim 20\%$ do not exceed a discharge of 35 m³/s. Another relevant result is that the surrogate model shows excellent performance in predicting extreme floods. For example, the largest flood, with a maximum discharge of $\sim 46$ m³/s, is predicted with a relative error that is lower than 1%.

In terms of efficiency and computational cost, the time required for building the model is highly-dependent on the number

10    of cases in the database. For the case of 49 scenarios, the time required for estimating the parameters is 4.78 seconds, which needs to be done only once, and we can later instantaneously predict the maximum water depth at the specific point we have chosen within the watershed.


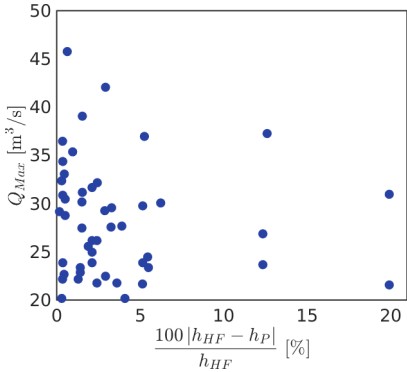

**Figure 9.** Results of the surrogate model for the flow discharge, compared to the data obtained from the high-fidelity approach.

### 4.3 Predicting the Flooded Area

With the objective of taking advantage of the high-resolution results from the database of high-fidelity simulations, we also explore the idea of predicting the total flooded area in the urban zone. We implement an individual surrogate model that predicts the maximum water depth at each point in the domain by using a first order regression function and a linear correlation function. To develop the surrogate model, we consider the same vector $X$ as input parameter, and the maximum water depth at each point of the domain as the output of the surrogate models. Then, we use in parallel the set of surrogate models for a unique set of input values, generating a map of the maximum flooded area, by using the maximum water depth predicted at each point of the domain.

We validate the model through the same cross-validation process described in Section 4.1, predicting the flood map for one case at a time, with a surrogate model based on a database that excludes the testing case. Due to the wide variety of storms that compose the database, result show areas that rarely are flooded are often predicted as flooded with water depths in the order of $10^{-2}$ cm. To deal with this effect, we filtered the predictions, considering flooded areas the zones with maximum water depth equal or larger than 10 cm.

We only assess the performance of this method as a qualitative error, as shown in Figure 10 for an specific storm with $X = [3.22\text{mm } 8.42\text{mm}^2 \ 0.45°\text{C } 29.05\%]$ and $Q_{Max} = 23.7$ m$^3$/s. In the figure, the green area represents nodes where the surrogate model and the high-fidelity simulation show equal flooding, the orange zone shows the nodes where the high-fidelity model shows flooding which are not predicted by the surrogate model, and conversely for the blue area, in which the surrogate model predicts flooding in sections that are dry from the high-fidelity model solutions.

Since this is a mountain river with a steep slope, the flooded area is usually very narrow along the main channel. The flooded nodes in this section are always correctly predicted by the surrogate model. In Figure 10, the over- and under-predicted regions exhibit water depths that are very close to the 10 cm threshold, which indicates that we could explore a better filter for future analysis.




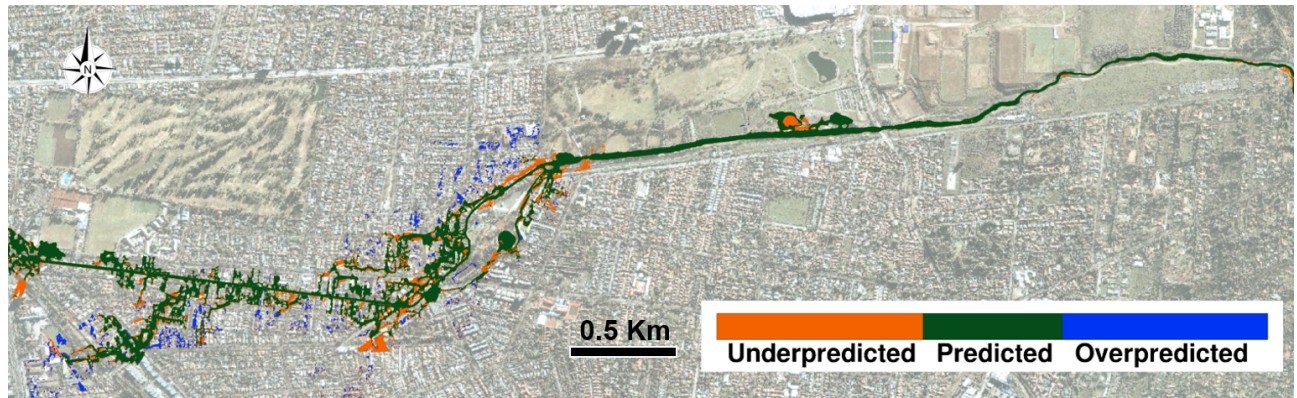

**Figure 10.** Qualitative error to assess the performance of a set of surrogate models working in parallel to predict flooded area in the study area. The results correspond to a randomly selected storm with input parameters: $P = 3.22$ mm, $M_{2_P} = 8.42$ mm$^2$, $T_{Min} = 0.45°$C, and $C = 29.05\%$. The green area is where the surrogate model predicts correctly the flooding, the orange shows zones where the surrogate model does not predict the flood, and the blue color represents zones where the surrogate model predicted flood, but the high-fidelity model does not. Background image from © Google Earth.

In terms of efficiency, this type of model to predict the total flooded area requires 2 hours to estimate the parameters with the current database of 49 cases. The number of outputs values is equivalent to the number of nodes of the discretization, which is equal to 2,728,058 nodes. It is important to note that this process needs to be done only once, then the prediction of the surrogate model for different scenarios is quasi-instantaneous.

## 5 Discussion

In this investigation we develop the first surrogate model applied to flood events in mountain regions. The results can help us identify some considerations for operational versions, as well as studying the performance of this approach for estimating the water depth and flooded areas in the city.

In terms of the implementation of the model, it is important to note that the accuracy of the results is highly related to the quality and size of the pre-computed database: Having a good representation of the physics of flood propagation to perform high-fidelity simulations determines the base error of all the predictions.

From the hydrological standpoint, we use a continuous rainfall-runoff model calculated from Hec-HMS, which is one-way coupled with the high-resolution two-dimensional hydrodynamic model to solve the propagation of the flood over the urbanized area. In this aspect, using field measurements to calibrate/validate both models, including additional physical phenomena that might have significant effects in other rivers such as the erosion/deposition of sediments, and having a better description of spatially-distributed parameters (for instance friction coefficients and land use changes, among others) can contribute to improve the accuracy of the high-fidelity models. In this sense, we must clarify that all the errors analyzed in this research are





only associated with the prediction of the surrogate model, but we do not include the error of the high-fidelity results, since it is unrealistic to have detailed measurements for every storm at each location of the city.

Another important step is building a database that is broad enough to cover the entire range of possible cases, and with a sufficient number of storms to provide a robust set of events for the interpolation method. Since the surrogate model is based

on Kriging, predicting cases outside the range of parameters simulated for the database can lead to results that are completely controlled by statistical errors. For operational purposes, a broader set of precomputed storms is required in order to have two different sets of data to cross-validate the surrogate model. However, since the record of precipitation in the study area is not long enough, synthetic storms might be required for regions with limited information.

To calibrate the surrogate model, we choose the best combination of regression/correlation functions based on the mean-

square error and the percent of cases predicted within one standard deviation of the prediction. The significant difference between the results for a zero or first order regression function versus the second order function shows that a linear function can capture the primary trend between inputs/outputs, and the correlation function provides the fluctuations from the structure of the statistical linear dependence. Over-constraining this relationship to a quadratic function results in a deterioration of the results of the surrogate approach. The different correlation functions seem to be more flexible to represent the noise associated

with the regression function.

To study the performance of the surrogate model to predict water depths at specific points in the watershed, we use a model built with a first-order regression function and a linear correlation function. From the results, we can clearly select events that are correctly predicted by the surrogate models in terms of relative and absolute error. They correspond to the $\sim 90\%$ of the predictions, with absolute and relative errors smaller than 0.25 m and $\sim 6\%$, respectively. On the other hand, we observe that

the remaining $\sim 10\%$ of the events are predicted with very significant errors, that reach 0.73 m or a $\sim 20\%$ relative error. We notice the association between low accuracy and high values of the standard deviation of the prediction. We use this relationship to apply a very simple filter that cleans up the results from the events that are not captured by the model. While not predicting all the events is a limitation for operative-oriented models, we consider a positive outcome that based on the value of the standard deviation we can determine if the prediction is correct. In this sense, more sophisticated filters can be applied to determine the

accuracy of the predictions, but our observations show that the model yields significantly larger standard deviations when the event is not correctly predicted.

We must also mention that the current database is built based on the historical record of events in the study area. This means the events are not uniformly distributed in the entire range of possible values for each input. In fact, the events that are the most represented in the database, have maximum flow rates at the critical point lower than 35 m$^3/s$, therefore they are not the most

destructive storms. For this reason, we study if specific values of the input parameters, or maximum flow rate at the prediction point might diminish the accuracy of the predicting, and we do not see that specific storms are consistently not predicted by the model. On the contrary, we highlight that the current database is able to predict frequent but also infrequent storms, with high values of $\overline{P}$, $M_{2_P}$, $T_{Min}$, and $Q_{Max}$. This shows that an uniformly populated database is recommended, but might not be required, as long as the entire range of possible cases is precomputed.





When trying to take advantage of the spatially-distributed information in the database, we predict flooded areas. While estimating the exact value of the maximum water depth at each point of the domain is the first approach, this presents new challenges since the prediction of each nodes does not consider the condition of surrounding points, discontinuities in the water surface might be predicted. In addition, this methodology does not accurately predict very shallow water depths. The wide variety of storms, influences the prediction of water depth at points that are rarely flooded. We solve this by applying a filter that neglects all the flooded zones with water depths shallower than 10 cm, which works for mountain regions where the slopes are steep as in the study case. However, flooded area predictions might be depended of this filter, especially in watershed characterized by very flat floodplains, and different approaches would be required. Thus, we only do a qualitative analysis to determine if the areas could potentially be flooded which is enough for quick-response systems. In terms of efficiency, this methodology provides the quasi-instantaneous response the surrogate models seek to provide.

In general terms, the main advantage of this statistical approach is to produce fast scenarios for decision makers, obtained directly from the characteristics of the meteorological event with no need of the high-fidelity approach, which requires running the computationally expensive hydrological and hydrodynamic models. While the accuracy of the prediction is affected by the statistical approach, the low errors of most of the prediction allow thinking on possible applications for quick-response systems, specially if a highly-populated database that covers the entire range of possible storms is used.

However, we aware of some of the limitations of this methodology. First, since the accuracy of the predictions if closely-related to the database, long term applications would need to include updates of the database, especially in the context of continuous modifications of the topography and land-use. Secondly, the replicability of this methodology to other watersheds might not be an easy task. While we can use the same interpolation methods in other watersheds, we would need to implement the high-fidelity models with the local data, and build a new database, which is by far the most time and computationally expensive step. Additionally, depending on the local physics of the study area, the inputs and outputs of the surrogate model ($X$ and $Y$) might need to be redefined as well.

## 6 Conclusions

The simulation of flood hazards by extreme precipitation in mountain streams requires numerical models capable of capturing complex flows that are influenced by the geomorphic features of the channel, and by high sediment concentrations that are common in these regions. In this investigation we develop two models for simulating the flow in an Andean watershed in central Chile: (1) A hydrologic model combined with a 2D hydrodynamic model that is coupled with the sediment concentration in mass and momentum; and (2) A surrogate model that employs pre-calculated scenarios of the previous models, to interpolate new cases using kriging.

With the combination of the hydrological and hydrodynamic models, we can capture the complexity of the flows, and estimate an accurate response to different storms. However, they require a very expensive computational resources and many CPU hours, which cannot be a tool suited for early warning systems. We use surrogate models to reduce the computational



cost, provide a fast prediction, and evaluate different scenarios to deal with the uncertainty of different future conditions, while preserving the effect of the physics of the flows in the prediction.

To develop this surrogate model, we characterize the extreme flood events by statistical parameters of the storm, and by the sediment concentration. We develop a database of 49 storms by using the high-fidelity hydrological and hydrodynamics

models. We perform kriging interpolation, based on the pre-computed database, to obtain water depth at a critical point in the domain, and flooded areas.

Results show a promising performance of the surrogate models for water depth prediction at a critical point in the domain. The current database can describe the flow propagation of floods, incorporating the connection of the hydrology and the complex hydrodynamics of the flow, highly affected by rapid slope variations and high sediment concentrations. Despite, this

has only the 49 storms measured in the entire historical record in the study area, the surrogate model can predict 90% of the storms with errors lower than 6% or 0.25 m. The remaining 10% of the storms are clearly not predicted, based on the high values of the standard deviation of the prediction. More importantly, the surrogate model shows good predictions for the most extreme events, with high values of mean precipitation, minimum temperature, storm duration and peak flow, which are sparsely represented in the current database. This shows that the most critical factor to design the database is covering the

entire spectrum of possible storms, rather than the uniformity of the storm representation in the database.

The implementation of surrogate models to predict flooded areas needs a more careful processing of the results. Since using individual surrogate models to predict maximum water depth at each point of the domain does not consider the influence of surrounding points, discontinuities in the water surface can emerge in the map. However, our results show that this is a powerful tool to qualitatively and quantitatively predict flooded areas, especially in mountain rivers where changes of the water depths

and the location of the wet/dry interface is more abrupt and easier to identify, compared to smoother topographies. For these other cases, we recommend using values of water depth in surrounding points as part of the inputs for a specific point.

In the future, we expect that this model will be incorporated in an automated framework, to provide an advanced tool for decision makers and stakeholders, who can evaluate scenarios without the technical expertise on the calculation of complex flows during extreme hydrometeorological events. However, some important issues need to be resolved: First, we recommend

increasing the high-fidelity database by using synthetic storms distributed with methods such as the latin hypercube sampling. This will improve the cross-validation process to determine the regression and correlation function, but it will also improve the accuracy of the predictions as the model will have more information to estimate cases that have not been pre-computed. In addition, we will test more sophisticated filters to improve the predictions and clean areas of the map with almost zero water depth, which also acquire great importance on flatter watersheds.

Other future topics of research include the incorporation of the time dependence, to predict water depths at different times during the storm events. This presents new challenges since the database is built with storms of different durations, so a normalization of the storm duration might be required.

*Competing interests.*  No competing interests are present.





*Acknowledgements.* This work has been supported by CONICYT/FONDAP grant 15110017, and by the Vice Chancellor of Research of the Pontificia Universidad Católica de Chile, through the Research Internationalization Grant, PUC1566 funded by MINEDUC.



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
