# Peer review of "Forecasting flood hazards in real-time: A surrogate model for hydrometeorological events in an Andean watershed"

_Natural Hazards and Earth System Sciences, 2019_

## Referee Comment (RC1) · Anonymous Referee #1 · 3 Mar 2020

General comments:

The authors develop a numerical tool for predicting flood hazard in real-time in an Andean watershed. The tool is based on a data-driven surrogate model of a physically-based hydrological-hydraulic modelling cascade. The topic is interesting and well-suited for NHESS.

The performance of the surrogate model in predicting water depth is however lower than I would have expected, and calls into question the application of this method in an operational version. As acknowledged by the authors, some storms are predicted with very significant errors. In my view, the paper falls short in explaining the reasons

behind this and the possible ways to improve the model performance. In section 4.2, the authors do not observe that the surrogate model does especially good or bad depending on the characteristics of the storms (understood here as the values of the 4 input parameters), but many other relevant aspects are not explored, such as: (1) the choice of input parameters itself, (2) the type of surrogate model, or (3) the number of events in the database. The later point is mentioned in the discussion (page 18, lines 9-11), but is not tested and shown in the results. Please find below some suggested readings that can provide insights into these questions:

Berkhahn, S., Fuchs, L., Neuweiler, I. (2019). An ensemble neural network model for real-time prediction of urban floods. Journal of Hydrology, 575, 743-754.

Bermudez, M., Cea, L., Puertas, J. (2019). A rapid flood inundation model for hazard mapping based on least squares support vector machine regression. Journal of Flood Risk Management, 12(S1), e12522

Jhong, B.C., Wang, J.H., & Lin, G.F. (2017). An integrated two-stage support vector machine approach to forecast inundation maps during typhoons. Journal of Hydrology, 547, 236– 252.

Razavi, S., Tolson, B. A., & Burn, D. H. (2012). Review of surrogate modeling in water resources. Water Resources Research, 48(7), W07401.

Even if depths are shallow, it is relevant to accurately predict flood extent for operational purposes and for extending the methodology to other sites (Page 17, lines 11-13 /Page 20, lines 5-7 of the manuscript). An additional step might be needed in the tool: a first binary classification model to predict when flooding occurs, and a second one to calculate its magnitude.

I suggest to evaluate the agreement between the surrogate and the physically-based depth maps obtained (ideally for all storms) by means of metrics such as the flood area index (a revision of commonly used metrics for this purpose can be found in: Stephens

et al. 2014. Problems with binary pattern measures for flood model evaluation. Journal of Hydrology 28 (18), 4928-4937). (Page 17, lines 14-18 / Page 21, line 19 of the manuscript).

Minor comments:

Page 4: Information on lines 9-10 seems to be repeated in lines 30-31.

Page 8: How are buildings represented in the mesh of the flood inundation model of the urban area? Is it a building block method?

Page 9, Lines 16-17. "...the zones where the sediment concentration produces significant changes on the velocity and flow depth of the flood" It's not clear to me how this is shown in Figure 5.

Page 14, line 22: Please write "event" in full for clarity.

Page 16, line 9: It would be useful to indicate the CPU time to simulate in the physically-based model, for comparison purposes.

Page 19, lines 21-22: Is this filter applied in this work?

Page 21, line 21: "we recommend using values of water depth in surrounding points as parts of the inputs for a specific point". As this possibility has not been tested in the paper, I suggest to remove this recommendation.

---

## Referee Comment (RC2) · Anonymous Referee #2 · 5 Mar 2020

The authors wrote about three objectives : -develop a"high-fidelity numerical model for inundation" - develop a surrogate model - apply this model for early warning. Apart the quick calculation and the selection of the inputs of the surrogate model, the third part is not developed enough for the reader to understand how this early warning system can operate. Particularly what is the chain from obtaining the input parameters for the surrogate model till the peak water depth at one building in the town? The first part that spreads over 7 pages is quite clear except the introduction of the sediment concentration and particularly, how is chosen the concentration for the 49 events of the data base used later on. Note that concentration cannot be selected independently from the event because for instance, high concentrations are specific from certain kinds of

events (snow melting or intense rainfall). Because the grid is sometimes very fine, one can guess that obstacles created by buildings, walls, etc can be taken into account but no word is written about the representation of any building or civil engineering structure. So it is difficult to conclude if it is really a "high fidelity numerical model". Part 2 is the core of the paper and takes more than 10 pages. The reader discovers while reading the paper that they are two surrogate models but only the first one is detailed and I could not understand how works the second one, which is not so important because this second model does not provide reliable results (I (of course) cannot understand why) and the authors should completely change the presentation fo this model if they wish to present it. For the first surrogate model, I understood that they use 48 set of parameters (different for each event) to obtain a mean forecast and a standard deviation. Because the procedure is not a standard one, I am not sure results can be trusted and I expect a comparison with a more standard procedure with, for instance, calibration on 25 events and evaluating results on the 24 other ones. Of course, it requires more time if you wish to test various sets of 25 events. Because I could not understand how concentration was determined for the 49 events, I could not judge the meaning of the results that concentration has no effect on the quality of the results. If the prediction is wrong (4 events out of 49), the authors do not provide what to do (if their model is used for early warning). §4.3 is an extension of the first surrogate model to other points and by this way to the flooded area but if the authors filter by a depth of 10 cm, I am wondering what means the flooded area (area with water depth higher than 10 cm?)and to which area of the finer model it is compared. The discussion (§5) is not structured and some new ideas are not clear (for instance sentence lines 7-8 of page 19 that is found again line 25 of page 21).

---

## Author Comment (AC1) · 13 May 2020

*Response to Referee #1*

We wish to thank the Reviewer for his/her thorough review of our manuscript and for the useful comments that helped us improve the quality of the paper. The specific issues raised by this Referee are addressed in detail below:

**General Comments**

*The authors develop a numerical tool for predicting flood hazards in real-time in an Andean watershed. The tool is based on a data-driven surrogate model of a physically-based hydrological-hydraulic modeling cascade. The topic is interesting and well-suited for NHESS.*
*The performance of the surrogate model in predicting water depth is, however, lower than I would have expected, and calls into question the application of this method in an operational version.*

We appreciate the positive comments from the Reviewer regarding the topic of our research. We would like to clarify this paper intends to show the first approach to use surrogate models as an alternative for water depth predictions in much shorter times than running high-resolution hydrologic and hydrodynamic models. While this idea can be used as the base of future operational models, we are aware that significant improvements and additional testing should be implemented in future research, as we have now discussed in the new version of the paper.
We apologize for not describing clearly the context of the study region in the original version of the manuscript, in which we should have explained the limitations of the surrogate model implementation in the Andes region. As we discussed in the companion paper recently published in NHESS (Contreras and Escauriaza, 2020), the main difficulty for predicting the flood propagation in the Andes is the very limited historical data and remarkably complex terrain. The Quebrada de Ramon watershed has only one water depth gauge with discontinuous 40 years of measurements and no rain gauges. Additionally, the magnitude of the floods makes measuring discharge even more difficult during storms.
In this case, we only have a record of 48 events with only one validation point in the entire region. With this limitation, we implemented a surrogate model that requires 4 inputs and predicts approximately 88% of the storms with a mean error of 2.22% at the validation point. Previous applications of surrogate models using similar methodologies for storm surge predictions have been built with 350 storms and 150 validation points, for 4 inputs. Their results have been validated with 20 storms with average errors of 4% and 3% for the significant wave height and water level respectively (see Taflanidis et al., 2013, for details).
While the order of magnitude of the average errors is similar, it is expected that a larger database to build our surrogate model would improve its accuracy, mainly because a wider variety of storms would be represented in the historical series.
*As acknowledged by the authors, some storms are predicted with very significant errors. In my view, the paper falls short in explaining the reasons behind this and the possible ways to improve the model performance.*

*In section 4.2, the authors do not observe that the surrogate model does especially good or bad depending on the characteristics of the storms (understood here as the values of the 4 input parameters), but many other relevant aspects are not explored, such as (1) the choice of input parameters itself, (2) the type of surrogate model, or (3) the number of events in the database. The latter point is mentioned in the discussion (page 18, lines 9-11), but is not tested and shown in the results. Please find below some suggested readings that can provide insights into these questions:*

*Berkhahn, S., Fuchs, L., Neuweiler, I. (2019). An ensemble neural network model for real-time prediction of urban floods. Journal of Hydrology, 575, 743-754.*

*Bermudez, M., Cea, L., Puertas, J. (2019). A rapid flood inundation model for hazard mapping based on least squares support vector machine regression. Journal of Flood Risk Management, 12(S1), e12522*

*Jhong, B.C., Wang, J.H., & Lin, G.F. (2017). An integrated two-stage support vector machine approach to forecast inundation maps during typhoons. Journal of Hydrology, 547, 236– 252.*

*Razavi, S., Tolson, B. A., & Burn, D. H. (2012). Review of surrogate modeling in water resources. Water Resources Research, 48(7), W07401.*

We sincerely thank the Referee for this comment, as we now realize that we did not explain thoroughly the relations and influence of the controlling factors of the extreme flood events in this region of the Andes mountains, and especially the process to select the input variables for the surrogate model, which is now explained in the new version of the paper.

Our previous research in these regions has shown that the variables that best explain daily discharges, particularly for low exceedance probabilities, are the cumulative precipitation over the previous 3 days and the minimum temperature on the day of the maximum discharge measured at a low elevation in the valley (Castro et al., 2019). This directly justifies the selection of the minimum temperature during the storm as part of the inputs, and underscores the importance of the liquid precipitation that occurs in the entire watershed during warm events (Contreras and Escauriaza, 2020).

The cumulative precipitation over the previous 3 days represents a combination of how much has rained and the duration of the event, which is also an indication of soil saturation in the watershed. Thus, we selected the mean of the precipitation during each event and the second moment of the distribution of precipitation as variables that represent the magnitude of the rainfall event and its distribution in time.

Finally, we consider the sediment concentration that might play a significant role on the flood propagation. Contreras and Escauriaza (2020) showed differences on the order of 25% for water depths calculated with clear water or 20% of sediment concentration. Additionally, differences of up to 0.5 m were observed in the urban area for hyperconcentrated flows. The main difficulty regarding the definition of this variable is the uncertainty of sediment concentration for each event, as localized landslides, previous recent storms, or interannual changes on the vegetation covering have produced different concentrations, which are also

difficult to measure in extreme flooding conditions. Therefore we selected this variable to evaluate potential scenarios, and analyze a flood with the same rainfall event, but under different sediment concentrations.

Magnitudes of sediment concentration have been reported for the largest flood registered in the watershed, which was generated during an abnormally warm storm, with periods of intense precipitation over partially saturated soils. The sediment concentrations during the events could not be directly measured, but it was estimated to be around ~40% (Sepulveda et al., 2006; Sepulveda and Rebolledo, 2008).

Regarding the types of surrogate models, we chose the methodology based on kriging due to the simplicity of its implementation in sites with limited information. To the best of our knowledge there are no previous studies that have developed surrogate models in ungauged mountain regions, therefore we selected a simple model, with the fewest number of parameters to calibrate, and keeping the physical meaning of all the inputs and parameters. We carried out a systematic study on the number of parameters, reducing and changing the combination of inputs, as shown in the following Table, and the model presented was the best in terms of reducing the error of the predictions.

| Mean Relative Error [%] | | | |
|---|---|---|---|
| Inputs | Regression Order 0 | Regression Order 1 | Regression Order 2 |
| P2TC | 1.93 - 3.55 | 2.26 - 2.70 | 12.61 - 20.81 |
| P2T | 2.53 - 3.74 | 2.36 - 3.05 | 5.75 - 9.73 |
| P2C | 2.39 - 3.52 | 2.51 - 3.26 | 6.64 - 7.96 |
| PTC | 2.61 - 3.53 | 1.99 - 2.49 | 3.68 - 4.19 |
| 2TC | 2.49 - 3.60 | 1.81 - 2.07 | 5.50 - 6.49 |
| P2 | 3.18 - 17.86 | 3.42 - 27.27 | 3.83 - 20.52 |
| PT | 2.39 - 4.41 | 2.38 - 3.77 | 2.24 - 3.87 |
| PC | 2.12 - 3.64 | 2.27 - 3.28 | 2.76 - 3.86 |
| 2T | 2.57 - 8.43 | 2.96 - 12.21 | 2.59 - 187.76 |
| 2C | 1.32 - 3.24 | 1.61 - 2.06 | 1.93 - 2.48 |
| TC | 1.78 - 3.20 | 2.21 - 2.90 | 2.55 - 3.40 |
| P | 4.14 - 1545.12 | 4.18 - 1561.64 | 4.02 - 1450.13 |
| 2 | 1.26 - 15.37 | 1.50 - 15.45 | 1.53 - 15.49 |
| T | 2.29 - 46445.19 | 2.55 - 46328.71 | 2.71 - 46322.26 |
| C | 1.83 - 33.37 | 2.20 - 37.27 | 2.19 - 34.89 |

| Cases Predicted within one standard deviation [%] | | | |
|---|---|---|---|
| | Regression Order 0 | Regression Order 1 | Regression Order 2 |
| P2TC | 77.55 - 89.80 | 75.51 - 89.80 | 85.71 - 91.84 |
| P2T | 53.06 - 67.35 | 53.06 - 71.43 | 53.06 - 75.51 |
| P2C | 69.39 - 85.71 | 67.35 - 83.67 | 67.35 - 83.67 |
| PTC | 63.27 - 75.51 | 69.39 - 71.43 | 75.51 - 81.63 |
| 2TC | 63.27 - 79.59 | 59.18 - 69.39 | 67.35 - 81.63 |
| P2 | 40.82 - 79.59 | 38.78 - 77.55 | 46.94 - 81.63 |
| PT | 46.94 - 65.31 | 46.94 - 71.43 | 46.94 - 67.35 |
| PC | 63.27 - 75.51 | 69.39 - 79.59 | 69.39 - 83.67 |
| 2T | 55.10 - 71.43 | 55.10 - 73.47 | 55.10 - 71.43 |
| 2C | 63.27 - 79.59 | 61.22 - 71.43 | 63.27 - 73.47 |
| TC | 69.39 - 85.71 | 69.39 - 81.63 | 69.39 - 81.63 |
| P | 48.98 - 67.35 | 48.98 - 67.35 | 53.06 - 65.31 |
| 2 | 32.65 - 71.43 | 30.61 - 75.51 | 34.69 - 77.55 |
| T | 44.90 - 67.35 | 44.90 - 65.31 | 44.90 - 63.27 |
| C | 61.22 - 83.67 | 57.14 - 81.63 | 61.22 - 83.67 |

*Even if depths are shallow, it is relevant to accurately predict flood extent for operational purposes and for extending the methodology to other sites (Page 17, lines 11-13 /Page 20, lines 5-7 of the manuscript). An additional step might be needed in the tool: a first binary classification model to predict when flooding occurs, and a second one to calculate its magnitude. I suggest evaluating the agreement between the surrogate and the physically-based depth maps obtained (ideally for all storms) by means of metrics such as the flood area index (a revision of commonly used metrics for this purpose can be found in Stephens et al. 2014. Problems with binary pattern measures for flood model evaluation. Journal of Hydrology 28 (18), 4928-4937). (Page 17, lines 14-18 / Page 21, line 19 of the manuscript).*

We agree with the Reviewer, as shallow depths are also relevant for the analysis of the flooded area, and the metrics proposed by Stephens et al. (2014) have provided information on this binary classification for a smaller number of nodes. However, the analysis for all the storms in the unsteady flows of our cases, including high spatial resolution that varies in size as we move further away from the channel, makes this specific analysis a formidable task, which is outside the scope of the present investigation. We initially focused on the development and validation of the surrogate model in a mountain region, with limited available data and based on kriging interpolation.

It is important to emphasize that the nodes that are far from the main channel do not get inundated very often, and they participate in a considerably smaller number of events within the database, which adds uncertainty and calls for a careful analysis on this specific point in future research.

To provide an analysis of the errors on shallow flooded areas in the present investigation, we have now performed an averaged analysis, comparing the outcomes of the surrogate model and the deterministic simulations, as shown in the following figure. In this case, we can identify in blue and red, specific regions that might show problems with the prediction of shallow inundations.

[Figure]

_**Minor comments:**_

*Page 4: Information on lines 9-10 seems to be repeated in lines 30-31. Page 8: How are buildings represented in the mesh of the flood inundation model of the urban area? Is it a building block method?*

**Response:**

We have improved the description on how we represent the vegetation and buildings in the hydrodynamic model. We modified the lines 9-10 and 30-31 in the new version of the paper, so the information is not repeated. We specify on line 31 that we use a bare Earth or digital terrain model for the hydrological model. On page 8, line 17, we describe how we use a digital surface model (DSM) for the hydrodynamic model, and represent the geometry of buildings in the urban area with their elevations in the computational grid.

*Page 9, Lines 16-17. "... .the zones where the sediment concentration produces significant changes on the velocity and flow depth of the flood" It's not clear to me how this is shown in Figure 5.*

**Response:**

We apologize for not explaining in detail the effects of sediment concentrations, as we described them above. We have modified the manuscript to incorporate the adequate details on how the sediment concentration modifies the hydrodynamics of the floods in the urban region, which is based on the companion paper (Contreras and Escauriaza, (2020),
 https://doi.org/10.5194/nhess-20-221-2020, 2020.

*Page 14, line 22: Please write "event" in full for clarity.*

**Response:**

Corrected.

*Page 16, line 9: It would be useful to indicate the CPU time to simulate in the physically-based model, for comparison purposes.*

**Response:**

In the new version of the manuscript we have incorporated the information about the elapsed times for the simulations on page 16 lines 12 and 13. This information highlights the importance of the fast response of the surrogate model, which is almost instantaneous, contrasted with 2 to 3 days of calculation for a high fidelity simulation.

*Page 19, lines 21-22: Is this filter applied in this work?*

**Response:**

Thanks for clarifying this point. We did not apply this filter to these results. All the errors and standard deviations are influenced by the incorrectly predicted events. We have modified the manuscript to recommend filtering the predictions based on large standard deviations. We observed that this procedure might be one way to identify predictions with large errors.

*Page 21, line 21: "we recommend using values of water depth in surrounding points as parts of the inputs for a specific point". As this possibility has not been tested in the paper, I suggest removing this recommendation.*

**Response:**
We deleted the recommendation, following this comment from the Reviewer.

**References:**

Taflanidis, A., Kennedy, A., Westerink, J., Smith, J., Cheung, K., Hope, M., and Tanaka, S.: Rapid assessment of wave and surge risk during
landfalling hurricanes: Probabilistic approach, Journal of Waterway, Port, Coastal, and Ocean Engineering, 139, 171–182, 2013

Castro, L., Gironás, G., Escauriaza, C., Barría, P., and Oberli, C.: Meteorological Characterization of Large Daily Flows
in a High-Relief Ungauged Basin Using Principal Component Analysis, Journal of Hydrologic Engineering, 24, 05019 027,
https://doi.org/10.1061/(ASCE)HE.1943-5584.0001852, 2019

Contreras, M. T., and Escauriaza, C.: Modeling the effects of sediment concentration on the propagation of flash floods in an Andean watershed, Natural Hazards and Earth System Sciences, 20, 221–241, https://doi.org/10.5194/nhess-20-221-2020, 2020.

Sepúlveda, S., Rebolledo, S., and Vargas, G.: Recent catastrophic debris flows in Chile: Geological hazard, climatic relationships, and human response, Quaternary International, 158, 83–95, https://doi.org/10.1016/j.quaint.2006.05.031, 2006.

Sepúlveda, S. and Padilla, C.: Rain-induced debris and mudflow triggering factors assessment in the Santiago cordilleran foothills, Central Chile, Nat Hazards, 47, 201–215, https://doi.org/https://doi.org/10.1007/s11069-007-9210-6, 2008

---

## Author Comment (AC2) · 13 May 2020

*Response to Referee #2*

We wish to thank the Reviewer for his/her thorough review of our manuscript and for the useful comments that helped us improve the quality of the paper. The specific issues raised by this Referee are addressed in detail below:

*The authors wrote about three objectives: -develop a"high-fidelity numerical model for inundation" - develop a surrogate model - apply this model for early warning. Apart from the quick calculation and the selection of the inputs of the surrogate model, the third part is not developed enough for the reader to understand how this early warning system can operate. Particularly what is the chain from obtaining the input parameters for the surrogate model till the peak water depth at one building in the town?*

**Response:**

We apologize because we failed to clarify the objectives of the present manuscript, as it is part of a comprehensive investigation of the basin, for which we had recently developed a high-fidelity model that we reported in the companion paper by Contreras and Escauriaza (2020).

The main goal of the current investigation is to develop and test the performance of a surrogate model, which can replace the computationally expensive combination of the hydrological and hydrodynamic models for flood predictions. While we do not mean to design a forecasting model in this research, we see the fast response and low computational costs of the surrogate model as valuable features, which can be used in operative forecasting systems and early warning systems.

We modified the manuscript to make the objectives clear. Also, we have specified that the idea of surrogate models could be used on early warning systems in areas with limited hydrometeorological data such as the Andes mountains, although further improvements will be required on future research.

*The first part that spreads over 7 pages is quite clear except the introduction of the sediment concentration and particularly, how is chosen the concentration for the 49 events of the database used later on. Note that concentration cannot be selected independently from the event because, for instance, high concentrations are specific from certain kinds of events (snow melting or intense rainfall).*

**Response:**

We also believe that the sediment concentration is related to the characteristics of the events, however, there is a significant uncertainty in these mountain environments to predict their values associated with specific hydrometeorological events:

Sediment concentration can play a significant role in the flood propagation. Contreras and Escauriaza (2020) showed differences on the order of 25% for water depths calculated with clear water or 20% of sediment concentration. Additionally, differences of up to 0.5 m were observed in the urban area for hyperconcentrated flows.  The main difficulty regarding the definition of this variable is the uncertainty of sediment concentration for each event, as

localized landslides, previous recent storms, or interannual changes on the vegetation covering have produced different concentrations, which are also difficult to measure in extreme flooding conditions. Therefore we selected this variable as independent to evaluate potential scenarios, and analyze a flood with the same rainfall event, but under different sediment concentrations.

Magnitudes of sediment concentration have been reported for the largest flood registered in the watershed, which was generated during an abnormally warm storm, with periods of intense precipitation over partially saturated soils. The sediment concentrations during the events could not be directly measured, but it was estimated to be around ~40% (Sepulveda et al., 2006; Sepulveda and Rebolledo, 2008).

In the present investigation, we chose values of sediment concentrations in a wide range between clearwater and 40%, through a Latin hypercube sampling. We selected this range based on previous the estimations, during the largest events in this watershed (Sepulveda et al., 2006). Additionally, our recent work in this river showed that the most significant differences in the hydrodynamics of the flow occur between 0 to 20% (Contreras and Escuariza, 2020). We are aware that this parameter might add an additional source of uncertainty, but one of the advantages of surrogate models is that they allow us to predict multiple possible scenarios, to deal with the uncertainty associated with the input values.

*Because the grid is sometimes very fine, one can guess that obstacles created by buildings, walls, etc can be taken into account but no word is written about the representation of any building or civil engineering structure. So it is difficult to conclude if it is really a "high fidelity numerical model".*
**Response:**
The representation of buildings is only included in the hydrodynamic model. Because we are using a digital surface model (DSM) to interpolate the topography with high resolution. The infrastructure is modeled with their higher elevation points in the computational grid.

*Part 2 is the core of the paper and takes more than 10 pages. The reader discovers while reading the paper that they are two surrogate models but only the first one is detailed and I could not understand how works the second one, which is not so important because this second model does not provide reliable results (I (of course) cannot understand why) and the authors should completely change the presentation fo this model if they wish to present it.*
**Response:**
We are sorry that we could not clarify this point in the original version of the manuscript. There is only one surrogate model, which is employed in two cases. First, we implemented a series of surrogate models by using multiple combinations of regressions and correlation functions. We evaluated the errors of the predictions and selected the combination of a first-order regression function and linear correlation function as the most accurate surrogate model for this watershed. Then we extended the same approach for the entire urban area by assuming that first-order

regression functions and linear correlation functions perform the best in all the cells of the domain.

*For the first surrogate model, I understood that they use 48 sets of parameters (different for each event) to obtain a mean forecast and a standard deviation. Because the procedure is not a standard one, I am not sure results can be trusted and I expect a comparison with a more standard procedure with, for instance, calibration on 25 events and evaluating results on the 24 other ones. Of course, it requires more time if you wish to test various sets of 25 events.*

**Response:**

As we discussed in the companion paper recently published in NHESS (Contreras and Escauriaza, 2020), the main difficulty for predicting the flood propagation in the Andes is the very limited historical data and remarkably complex terrain. The Quebrada de Ramon watershed has only one water depth gauge with discontinuous 40 years of measurements and no rain gauges. Additionally, the magnitude of the floods makes measuring discharge even more difficult during storms.

In this case, we only have a record of 48 events with only one validation point in the entire region. With this limitation, we implemented a surrogate model that requires 4 inputs and predicts approximately 88% of the storms with a mean error of 2.22% at the validation point. Previous applications of surrogate models using similar methodologies for storm surge predictions have been built with 350 storms and 150 validation points, for 4 inputs. Their results have been validated with 20 storms with average errors of 4% and 3% for the significant wave height and water level respectively (see Taflanidis et al., 2013, for details).

While the order of magnitude of the average errors is similar, it is expected that a larger database to build our surrogate model would improve its accuracy, mainly because a wider variety of storms would be represented in the historical series.

In this case, the synthetic series obtained from the hydrological model are based on the limited data of the only discharge gauge available in the entire basin. Additional simulations do not add statistically meaningful information that can improve the predictions, since they are based on the same set of data. To the best of our knowledge there are no previous studies that have developed surrogate models in almost ungauged mountain regions, and this is the first step to implement these tools in the future.

*Because I could not understand how concentration was determined for the 49 events, I could not judge the meaning of the results that concentration has no effect on the quality of the results. If the prediction is wrong (4 events out of 49), the authors do not provide what to do (if their model is used for early warning).*

**Response:**

As we explained in previous comments the sediment concentration is randomly selected in a range of values that have been estimated for large floods in the region. While we tried to find a relationship between the values of the inputs and the wrong predictions of the surrogate model, the model does not perform particularly badly for specific types of storm. However, we recommend considering large standard deviations of the predictions as a filter for incorrect predictions, as we mention in the new version of the paper.

For early warning systems, it would be desirable to use a larger database that includes a wider variety of storms and more densely sampled, which is not available in this case. Literature shows that increasing the number of high fidelity simulations to the order of thousands should improve the performance of the model, but the limitation in this case is the hydrometerological information to create the scenarios. In the real implementation of an early warning system, the inputs have uncertainty that the surrogate model can help to deal with, by assessing multiple scenarios for a single storm. This also allows discriminating the rare scenarios when the surrogate model is not able to correctly predict the flooding area.

*§4.3 is an extension of the first surrogate model to other points and by this way to the flooded area but if the authors filter by a depth of 10 cm, I am wondering what means the flooded area (area with water depth higher than 10 cm?)and to which area of the finer model it is compared.*

**Response:**

Since we applied the 10 cm minimum water depth for the results of the surrogate model, we compared the results with the high-fidelity simulation after the same filter.

*The discussion (§5) is not structured and some new ideas are not clear (for instance sentence lines 7-8 of page 19 that is found again line 25 of page 21).*

**Response:**

We have reformulated the discussion to improve the text and clarify the main ideas developed in the manuscript. We thank the Referee for the thorough review of our work.

---

## Referee Report (RR1)

The second version of the paper is clearer than the first one. However, some additional explanations are required.

Page 3 line 11, there is the word "outputs" while there is only one output. Please correct if necessary or explain.

Page 3 line 14: what are the differences between the meta-model and the surrogate model?

The figure 4 is not clear enough.

Page 13 lines 17-18 and later on: define (for instance by an equation and at least by a variable name), MSE, PWSD and above all standard deviation because you use «standard deviation » several times in the paper and it is not clear if if it is the standard deviation of the same samples and on the same variables. Particularly, "standard deviation" on Figure 7 cannot be understood because there is only one prediction for each event.

Comparison of the surrogate model against observations is expected either in §4.2 or §4.3.

Page 17 the sentence of lines 21 and 22 cannot be understood. You should detail the explanation.

At the end of §4.3, I expected an estimation of the error. For instance, the number of points predicted over the total of underpredicted + predicted + overpredicted. I guess the number is so low that you do not wish to show it. The information that the model provides poor results is nevertheless an interesting information.

Page 19 line 32:  as previously, what means « standard deviation of the prediction »: the surrogate model provides one value of one output for one event.

Page 20 line 1: « standard  deviations »:  similar remark as previous one

Page 22 line 2: you recommend but have no proof that it would be an efficient method: so cancel it.

---

## Author Response (AR2)

***Response to Referee #1***

***The second version of the paper is clearer than the first one. However, some additional explanations are required, particularly to indicate how are calculated the standard deviations. Efficiency of the flooded area calculation and comparison of surrogate model to observations should be provided.***
**Response:**
We kindly thank the Referee for the effort of reviewing our manuscript a second time. We have addressed these minor issues in the new version of the paper, and we have also answered the details below.

**Minor comments:**

***Page 3 line 11, there is the word "outputs" while there is only one output. Please correct if necessary or explain.***
**Response:**
Corrected.

***Page 3 line 14: what are the differences between the meta-model and the surrogate model?***
**Response:**
Both names are employed to describe the statistical approaches built on the physics-based models, which are designed to provide flooding predictions based on simpler rules.
In the new version of the manuscript we have modified the text in page 2 line 31, where we introduced the concepts to explain that they refer to the same modeling approach.

***The figure 4 is not clear enough.***
**Response:**
We updated figures 3 and 4 to improve the quality of the image and size of the fonts.

***Page 13 lines 17-18 and later on: define (for instance by an equation and at least by a variable name), MSE, PWSD and above all standard deviation because you use «standard deviation » several times in the paper and it is not clear if if it is the standard deviation of the same samples and on the same variables. Particularly, "standard deviation" on Figure 7 cannot be understood because there is only one prediction for each event.***
**Response:**
We thank the Reviewer for pointing out this issue, since clarifying the use of the mean and standard deviation is critical to explain the performance analysis that we carry out in the paper. These values refer to the mean and deviation of the predictions, which are defined in equation 9. We included some comments on page 13, lines 4-5 and 18-19, to explain their definition and relations with the surrogate model.

***Comparison of the surrogate model against observations is expected either in §4.2 or §4.3.***
**Response:**
One of the main limitations of implementing surrogate models in the Andes regions is the very limited historical data. The *Quebrada de Ramon* watershed has only one water depth gauge with discontinuous 40 years of measurements. Since we need a dense cloud of points to implement the surrogate model, we had to use the high-fidelity simulations as a benchmark for the surrogate model. We acknowledge that the high-fidelity model can introduce some errors, but we tried to further minimize them by also performing a comprehensive validation study with analytical solutions and detailed experiments (see Guerra et al., 2014 and Contreras & Escauriaza, 2020, for details). Performing new measurements to produce more observations is out of the scope of this paper at the moment, especially in this site with sporadic but strong events.

***Page 17 the sentence of lines 21 and 22 cannot be understood. You should detail the explanation.***
**Response:**
We corrected the line and improved the description of the methodology.

***At the end of §4.3, I expected an estimation of the error. For instance, the number of points predicted over the total of underpredicted + predicted + overpredicted. I guess the number is so low that you do not wish to show it. The information that the model provides poor results is nevertheless an interesting information.***
**Response:**
We limited this analysis, due to space considerations and specific objectives of the paper, it was not an attempt to conceal the results. We have now incorporated the values of three other measurements in the manuscript to quantitatively evaluate the model's performance. Please refer to page 18, lines 13-19.

***Page 19 line 32: as previously, what means « standard deviation of the prediction »: the surrogate model provides one value of one output for one event.***
**Response:**
Corrected.

***Page 20 line 1: « standard deviations »: similar remark as previous one***
**Response:**
Corrected.

***Page 22 line 2: you recommend but have no proof that it would be an efficient method: so cancel it.***
**Response:**
We have deleted that point in the last version of the manuscript.

**General Comments**

*I would like to thank the authors for responding to my previous comments, and clarifying some of the issues encountered during the previous review. However, it seems that only minor changes have been introduced into the revised version of the manuscript, hence a number of my previous comments still stand.*

*Accepting that the performance of the surrogate model might be in line with expectations, it seems weird to refer to a single study (Taflanidis et al., 2013) dealing with hurricane wave and surge risk (instead of precipitation-driven flooding) for comparison. This seems to be the only reference in the paper to previous works that have developed surrogate models for similar purposes. In my previous review I provided some references which might be better suited. At present there is also no reference in the manuscript to the error values reported in previous studies to frame the expectations of model performance.*

*The authors conclude that "our results show that this is a powerful tool to qualitatively and quantitatively predict flooded areas…". However, if I understood correctly, they have not evaluated the agreement between the surrogate and the physically-based depth maps obtained, but for a single storm. I understand the limitations encountered to produce accurate flood depth maps (discontinuities in the water surface, shallow water depths…), but the quality of the maximum flood extent predictions (flooded vs dry areas) considering water depths above a certain threshold (e.g., the 10 cm indicated by the authors) should be compared systematically and quantitatively for all storms. In my previous revision I suggested the flood area index as a possible metric.*
**Response:**

We apologize if we did not address the previous comments properly. We appreciate the suggestions of the references as they are appropriate examples, however, while they have similar objectives, there are significant differences on the application of the surrogate models and physical conditions compared to what we find in the Andes region.

In the new version of the manuscript we have incorporated the references by Jhong et al. (2017) and Bermúdez et al. (2019) as examples of implementations of surrogate models for flood hazard mapping in zones with milder slopes, and models of typhoons at larger scales. We have now incorporated comments on these references in the introduction section (see page 2 lines 32-33).

We have also added a comparison with the results obtained by Bermúdez et al. (2019) in section 4.2 (see page 14 lines 17-19 to page 15 lines 1-4). We note that the mean absolute error in Bermudez's work compared with a hydrodynamic model is 2.3 cm, while in our model is 13 cm. However, we point out a couple of important differences:

Bermudez et al. (2019) used as input parameters the peak discharge of the inlet hydrograph and the water level at the channel outlet. This means their surrogate model only replaces a hydrodynamic propagation model. In our research, however, we seek to use the hydrometeorological data before the events to predict the water levels, because this is the only information available before the storms. Thus, the surrogate model that we implemented intends to replace the combination of hydrological and hydrodynamic simulations, considering the effects of hyperconcentrated flows, and including the meteorological and terrain variables that define the magnitude of the event, which can be easily obtained.

Secondly, we implement the model with a dataset of 49 storms, all with different hydrometeorological conditions. Bermúdez et al. (2019), on the other hand, implemented the surrogate model with 100 storms based on 15 historical rainfall events and variations of the abstraction and potential infiltration in the hydrological model. This means that we are trying to implement a model that covers a much wider spectrum of storms with half of the simulations, which logically would lead to larger errors on the predictions.

We would also like to clarify that we used all the available storms to implement and validate the flooded area models. The map presented in the manuscript is an example of the results we obtained from the surrogate model, and how the errors are spatially distributed.

Considering the Referee's suggestion, we have included a more quantitative analysis of the model's performance. We have followed the guidelines provided in Stephens et al. (2014) to calculate the hit rate, false alarm rate, and the critical success index. Please refer to page 18, lines 13-19, for more details in this regard.

[revised manuscript text omitted]